# Perceptions towards the COVID-19 Pandemic during Different Lockdown Levels among International Students in Taiwan

**DOI:** 10.3390/ijerph20064944

**Published:** 2023-03-11

**Authors:** Yi-Hao Weng, Wei-Ting Chien, Felix Trejos Prado, Chun-Yuh Yang, Hung-Yi Chiou, Wei-Cheng Lo, Chung Bui, Ya-Wen Chiu

**Affiliations:** 1Department of Pediatrics, Chang Gung Memorial Hospital, Chang Gung University College of Medicine, Taipei 10507, Taiwan; yihaoweng@cgmh.org.tw; 2Master Program in Global Health and Development, College of Public Health, Taipei Medical University, Taipei 11031, Taiwan; shaba1002@gmail.com (W.-T.C.); ftrejosp@yahoo.com (F.T.P.); hychiou@tmu.edu.tw (H.-Y.C.); 3Department of Public Health, Kaohsiung Medical University, Kaohsiung 80708, Taiwan; chunyuh@kmu.edu.tw; 4Master Program in Applied Epidemiology, College of Public Health, Taipei Medical University, Taipei 11031, Taiwan; nicholaslo@tmu.edu.tw; 5Institute of Population Sciences, National Health Research Institutes, Miaoli 35053, Taiwan; 6Department of Health Communication and Education, Quang Ninh Provincial Center for Disease Control, Ha Long 01060, Quang Ninh, Vietnam; chungbuiytcc@gmail.com

**Keywords:** COVID-19, international student, lockdown, knowledge, attitude, practice

## Abstract

International students face many impediments under the COVID-19 pandemic. The objectives of this study are to assess the association between the perceptions of international students and the lockdown policy for COVID-19. In 2021, three different levels of lockdown policy were enforced, including level I from January to April, level III from May to July, and level II from August to December. We conducted three surveys for international graduate students using a validated questionnaire during the different lockdown levels. We collected 185, 119, and 83 valid questionnaires in level I, II, and III, respectively. There were linear trends in the correlations of lockdown policy with the knowledge (*p* = 0.052), attitudes (*p* = 0.002), and practices (*p* < 0.001) of COVID-19. In brief, the stricter the lockdown policy, the better the students adhered to sufficient knowledge, positive attitudes, and healthy practices. Furthermore, there were significant linear correlations of lockdown policy with the transportation, school study, leisure, family life, and diet behavior. In conclusion, lockdown policy had important impacts on the knowledge, attitudes, practices, and daily lives of international students. The findings indicated that the lockdown system and its corresponding measures appear to affect perceptions in a positive way.

## 1. Introduction

COVID-19 has been a global threat around the world for over three years [1,2]. The massive spread of COVID-19 not only had a public health impact but also led to a disruption to the education system [3,4,5,6]. Lockdown, one of the social isolation restrictions, is effective to prevent the spread of COVID-19 pandemic [1]. In Taiwan, the Central Epidemic Command Center (CECC) formulated a warning system of lockdown policy for the COVID-19 pandemic (Table 1). This policy aimed to clean all possible infectious sources to reach zero-COVID from February 2020 to February 2022. From the beginning of 2020 through April of 2021 (level I), the confirmed cases were only a few in Taiwan. Thus, the policy for COVID-19 only included quarantine for suspected cases and wearing a mask indoors. However, there was an increasing number of cases during May to July 2021 (level III). A very strict policy for COVID-19 limited the life of all people in Taiwan. With a better control from the end of July 2021 until February 2022 (level II), the policy became looser than level III, with partial restrictions for almost every individual in Taiwan, including international students. In fact, these measures proved to affect international students more due to the culture differences and the language barrier.

International students often face complicated challenges such as psychological and sociocultural adjustment [7,8,9,10]. The sector of society that is pursuing a degree abroad is more vulnerable to the stress caused by the COVID-19 pandemic [11,12]. Thus, they require more attention. First, international students are at greater risk of COVID-19 because of their high number of social activities. They experience the psychological impact of COVID-19 and higher levels of stress, anxiety, and depression [12]. Compared to local students, international students face even more impediments under the COVID-19 pandemic and beyond. Existing stress related to the acculturation demands of studying abroad may be amplified during the pandemic [13,14]. Whether to return to their home country or stay in the host country and wait for the end of the outbreak becomes a dilemma. Those who remain in their host countries face unmet psychological needs concerning loneliness as a result of their physical separation from their loved ones and a loss of social support in the local culture—not to mention the psychosocial issues involved with the wider society’s response to COVID-19.

The perceptions of international students toward the COVID-19 pandemic have been surveyed in many countries [12,14,15,16,17,18,19,20,21,22,23]. Despite the discrepancy in the government policy and disease burden across the countries in these studies, all the findings indicated that international students confronted various aspects of mental problems [24,25]. However, surveys analyzing the correlation of lockdown policy with students’ perceptions towards COVID-19 are scant. Thus, the current study aims to investigate the perceptions of international students during different levels of lockdown and further determine how the perceptions of international students change with different lockdown levels for COVID-19. Our findings provide critical evidence of the impacts of lockdown policy on the thoughts and behaviors of international students during the COVID-19 pandemic.

## 2. Materials and Methods

### 2.1. Study Design

This study was conducted using an online questionnaire survey system from January 2021 to August 2022 in Taiwan. The disease burden and lockdown policy of COVID-19 during the study period are shown in Table 1. Three cross-sectional questionnaire surveys of international students in Taipei Medical University were conducted, including the first survey between 3 January and 30 March, second survey between 1 and 16 July, and third survey between 4 and 20 August 2021. International students at Taipei Medical University were eligible for enrollment. Exchange students, native speakers of Chinese, and those who did not stay in Taiwan or study at Taipei Medical University during the study period were excluded. Potential participants were contacted through the Office of International Engagement at Taipei Medical University. We informed them about our study and asked for their assistance to distribute the questionnaire link and/or QR code to the international students.

### 2.2. Questionnaire Survey

The questionnaire surveys were carried out via SurveyCake, an online questionnaire survey tool. The structured questionnaire was developed using questions based on previously conducted surveys [26,27,28,29] and modified by five experts with backgrounds in medicine, statistics, and public health. All questions were developed in English. Questions included demographic information and items for measuring the knowledge, attitudes, practices (KAP), impacts, and information sources of the COVID-19 pandemic (see Appendix A). Demographic information included gender, age, home country, ethnicity, religion, marital status, pursuing degree, living arrangement, and stay duration in Taiwan. Impacts on daily life included transportation (commute), study, leisure, family life, diet behavior, and income. Questions were rated using a Likert 5-point scale (strongly agree, agree, neutral, disagree, and strongly disagree) for knowledge, beliefs, and attitudes. For the practices and impact of COVID-19, questions were also rated using a Likert 5-point scale (never, seldom, sometimes, often, and always).

### 2.3. Ethical Consideration

Our study protocol was approved by the Institutional Review Board of Taipei Medical University (number N202008053). The questionnaire was accompanied by an introductory letter stating the purpose of the study and promising confidentiality. Informed consent was obtained from participants before they started answering the questions.

### 2.4. Validity and Reliability of Questionnaire

To assess the clarity and ease of completion, the content validity was examined by five experts with more than 20 years each of professional service in public health, global health, infectious disease, and clinical medicine. In addition, test-retest reliability coefficients were used to estimate the internal consistency of all indices. The same questionnaire was pilot tested twice at an interval of 1 week on a sample of 16 international students. A content validity index of 0.98 and reliability coefficient of 0.83 indicated sufficient validity and reliability of parameters in the questionnaire survey.

### 2.5. Statistical Analyses

The Likert 5-point scale was dichotomized for further analyses. A self-rating report of either “strongly agree” or “agree” was regarded as a favorable answer, while the other three (“neutral”, “disagree”, and “strongly disagree”) were viewed as unfavorable answers for the beliefs and attitudes towards COVID-19. As for the practices and impacts towards COVID-19, a self-rating report of either “often” or “always” was regarded as a favorable answer, while the other three (“never”, “seldom”, and “sometimes”) were viewed as unfavorable answers. Sufficient knowledge was defined as correct answers for all questions. Positive attitudes and healthy practices were defined as all questions regarding COVID-19 were favorable responses.

A commercially available program was used for all statistical analyses (SPSS version 19.0 for Windows, Chicago, IL, USA). To analyze categorical variables, we used a chi-square test or Fisher’s exact test when appropriate. Mantel–Haenszel test was used to determine linear association. Significance was defined as *p* < 0.05 in two-tailed tests.

## 3. Results

### 3.1. Demographic Data of Participants

During the first survey (from January to March 2021) in level I lockdown, 192 participants completed the questionnaire. We excluded 7 invalid participants, leaving 185 participants for analysis. In this period, the number of confirmed and death cases was significantly lower than those in most countries. Thus, the daily lives were not restricted at all in Taiwan. Nevertheless, wearing a mask was mandatory indoors in public areas. From the beginning of the pandemic to this period, the Taiwan government used strict border controls, such as closing the gates of departure and entry, and prompt tracing systems to manage COVID-19 pandemic.

Since April 2021, the number of confirmed cases increased due to the spread of pilots of international flights. Thereafter, the lockdown level was raised to III from 15 May 2021 to 26 July in Taipei. At the second survey (July 2021), in level III lockdown, 87 participants completed the questionnaire. We excluded 4 invalid participants, leaving 83 participants for analysis. In this period, the number of confirmed and death cases was more than 10 per day (the maximal numbers of confirmed cases were less than 500 per day). Nevertheless, the confirmed cases were still lower than those in most countries. International students were not allowed to go to their universities. Lectures were conducted via online systems.

With the persistent rigorous controls, the number of confirmed cases decreased to less than 100 from the beginning of July onward. At the end of July, the disease burden declined, and the CECC announced the lockdown turned to level II. For the third survey (August 2021), in level II lockdown, 256 participants completed the questionnaire. We excluded 137 invalid participants, leaving 119 participants for analysis. During this period, lectures for international students were hybrid, including online and in the classroom with a limited number of students. Although the government used different measures for people living in Taiwan during the study period of January to August 2021, the policy for border control was approached with the same strictness. For example, a 14-day quarantine was mandatory for all confirmed cases and people entering Taiwan during the whole study period.

The demographic information of participants is shown in Table 2. There was no significant difference in the demographic information among three surveys, including gender, age, home country, ethnicity, religion, marital status, pursuing degree, living arrangement, and stay duration in Taiwan.

Female participants were greater in. number than males. Approximately 60% of participants were aged ≥30 years old. About one-third of participants were Indonesian, and one-third of participants were Vietnamese. Over 80% of participants were Asian. There was diversity in religion, such as Islam, Christianity, and Buddhism. More than one-half of participants were single. Nevertheless, about two-thirds of participants did not live alone. In addition, about one-third of participants pursued a master’s degree, and two-thirds of participants pursued a doctorate degree. Over 60% of participants had been in Taiwan more than one year.

### 3.2. Knowledge, Attitudes, and Practices of International Students

Figure 1 demonstrates the knowledge, attitudes, and practices of COVID-19 in the three different lockdown levels. There were significant differences in the attitudes and practices among the three lockdown levels (*p* = 0.001). In addition, a linear correlation between lockdown policy and the attitudes (*p* = 0.002) and practices (*p* < 0.001) of international students was noted—the stricter the lockdown policy, the better the international students adhered to positive attitudes and healthy practices. Furthermore, there was a linear trend in knowledge among the three surveys (*p* = 0.052).

### 3.3. Impacts of Lockdown Policy

The impacts of lockdown policy on daily lives are shown in Table 3. In lockdown level I, the most common impact for international students was transportation, followed by leisure, study, income, family life, and diet behavior. However, the most common impact in lockdown level II was leisure, followed by study, transportation, family life, diet behavior, and income. As for lockdown level III, the most common impact was study, followed by leisure, transportation, family life, diet behavior, and income. More than one-half of international students suffered significant impacts on transportation, study, leisure, family life, and diet behavior in level III.

There were significant changes in transportation, study, leisure, family life, and diet behavior among the three lockdown levels—the stricter the lockdown policy, the greater the influence on the perceptions of international students. As for income, there was no significant change among the three lockdown levels.

### 3.4. Information Source of COVID-19

Figure 2 shows the information source from which students learned about COVID-19. There was a significant difference in the numbers regarding searching for COVID-19 information among the five sources—social media, government channels, friends/neighbors/relatives, and printed media (*p* < 0.001). The most commonly accessed source was social media, followed by government channels, friends/neighbors/relatives, and printed media among three levels of lockdown policy. As for the option “others”, the most common source was private media from hometown, followed by private media from Taiwan and hearsay.

The most reliable information source is shown in Figure 3. There was a significant difference in the belief of the most reliable information among the five sources—the Taiwan government, research results and scientific journals, the World Health Organization (WHO), Taipei Medical University, and friends/neighbors/relatives (*p* < 0.001). The most reliable source was the Taiwan government, followed by research results and scientific journals. Friends/neighbors/relatives were the least reliable information source. The option “others” included private media from hometown, private media from Taiwan, and hearsay.

## 4. Discussion

In this study, we examined the correlation of COVID-19 policies with the perceptions of international students by conducting three surveys during three different lockdown levels. This questionnaire study allowed us to compare and contrast various levels of perceptions between different levels of the COVID-19 lockdown. The findings indicated that their compliance towards the COVID-19 pandemic was associated with the different levels of lockdown. To our knowledge, this is the first study to investigate how international students perceived and behaved according to the lockdown system.

The COVID-19 pandemic has brought the world to a relative stall. Unprecedented measures have been adopted worldwide to control the rapid spread of the ongoing COVID-19 epidemic. Despite these measures, the infection has been uncontainable in many countries due to different factors including people’s KAP towards this disease. KAP surveys are commonly used to identify knowledge gaps and behavioral patterns among sociodemographic subgroups to implement effective public health interventions [30]. KAP empirical studies can expose fundamental information to determine the types of measures that can positively affect an infectious disease outbreak [31]. Sufficient knowledge can change attitudes and then change practices. Thus, improving KAP is a potentially valuable tactic for insight into addressing misunderstandings [32]. Evidence of an unequal burden of COVID-19 is also emerging fast. People living in impoverished and racially and economically polarized areas showed considerably more significant morbidity and mortality rates of COVID-19. Thus, KAP towards COVID-19 plays an important role in assessing the willingness of a community to adopt behavioral change initiatives during the COVID-19 pandemic [33,34,35]. Improving KAP is a potential valuable tactic for insight into addressing misunderstandings among all the sectors of society, including those in vulnerable situations, such as the international students. Nevertheless, surveys focusing on the KAP of international students are limited [35].

In this study, we verified significant discrepancies in KAP among the different levels of lockdown policy. First, there was an increasing trend of sufficient knowledge of COVID-19 with the rise of lockdown level. We originally assumed that the rise of knowledge at level III would remain high as it downgraded back to level II. Contrary to what we expected, the knowledge level dropped significantly. The findings suggest that repetition and review are important when it comes to COVID-19-related health education. Second, positive attitudes of COVID-19 were significantly higher in lockdown level II and III compared with level I. Third, good practices toward COVID-19 proportionally increased with the rise of lockdown level. In general, good levels of knowledge were positively associated with optimistic attitudes and appropriate practices. Our study showed similar findings in international students. These results lead to the conclusion that lockdown policies had significant impacts on their KAP.

In addition to KAP, our study demonstrated that lockdown policies affected the daily lives of international students. Many reports showed that physical activity decreased, and psychosocial distress increased during the lockdown period for students [15,22,24,36]. Our investigation further extended their inquiry by showing a linear correlation of the impacts with different lockdown levels. In addition, it is not surprising to see that the lockdown policies affected their study, transportation, and diet behavior because these activities were restricted in high lockdown levels. For example, the diet behaviors changed greatly because dining at restaurants was totally prohibited during lockdown level III.

Our study identified the main sources of information of COVID-19 among international students. These findings are useful to identify misinformation and plan strategies to deal with it. In our survey, the support of the university was regarded as one of the positive attitudes. The results are in accordance with previous reports showing that universities play important roles to support students’ psychological stress during the COVID-19 pandemic [19,23]. Universities should provide sufficient support to reduce the anxiety of international students [37]. In addition, the government and universities encourage them to implement necessary preventive measures in Taiwan. Our results showed that the government is the most reliable information source, making the lockdown system and its corresponding measures more effective in affecting their perceptions. In Taiwan, the information for COVID-19 has spread instantly and transparently through mass media such as social media, television, and schools [38]. The surveillance of the Taiwan government was most reliable not only for international students but also for foreigners living in Taiwan [39]. Furthermore, we observed that the WHO was not an important reliable resource during the COVID-19 pandemic. This is probably because the WHO could not provide instant information about Taiwan. However, it is of interest to note the increased reliance on the WHO and the decreased reliance on the Taiwan government in the survey during the level II lockdown. We speculate that the process of a downgrade from lockdown level III to level II may cause a degree of untrustworthiness from the perspective of international students.

In our study, international students coped with the lockdown policies well. Their adherence seemed not to be affected by cultural diversity and the linguistic barrier. First, international students were able to search for up-to-date information for COVID-19 [38]. Clear instructions provide sufficient information for international students to cope with the COVID-19 pandemic. Second, the impediments of language and culture were minimal because of their high level of education. Foreigners with lower educational levels need more educational intervention than those with higher educational levels [40]. Third, as the period of this study was during the second year of lockdown in the world, international students should understand COVID-19 better than in 2020 [41]. Taken together, we suggest that education is helpful to overcome the culture and language impediments for international students.

With respect to limitations, the findings of our study should be interpreted cautiously for several reasons. First, this is a self-report survey and not an audit of actual practice. The results may not reflect the realities of practice under everyday conditions. We cannot be sure that these self-reported changes were fully translated into the general population. Second, as our survey was not linked to individual students, the respondents in the first survey did not overlap completely with those who completed the other questionnaire surveys. We believe the influence was minimal because of the high similarity of demographic backgrounds among the three surveys. In addition, we excluded language learners (studying at Mandarin training centers) and exchange students because they stay in Taiwan for a shorter period. Our study enrolled international students who stayed in Taiwan from January to August 2021. Therefore, those who stayed for a short duration were not eligible for enrollment. Third, this study was based on a private university; thus, the results may not be generalizable to all international students worldwide.

## 5. Conclusions

In this study, we used the same tool to conduct the questionnaire survey during three different lockdown periods. The lockdown policy in Taiwan affected the KAP and daily lives of international students. The findings indicated that the lockdown system and its corresponding measures appear to affect their perceptions in a positive way. Our results provide valuable information regarding how international students perceived and practiced their preventive measures for the COVID-19 pandemic. The data can help the government and universities to develop effective interventions to support international students.

## Figures and Tables

**Figure 1 ijerph-20-04944-f001:**
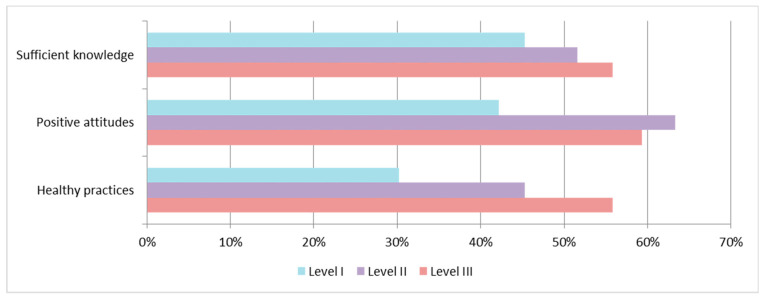
Knowledge, attitudes, and practices of COVID-19 in the three different lockdown levels. Level I, not restricted; level II, partially restricted; level III, restricted.

**Figure 2 ijerph-20-04944-f002:**
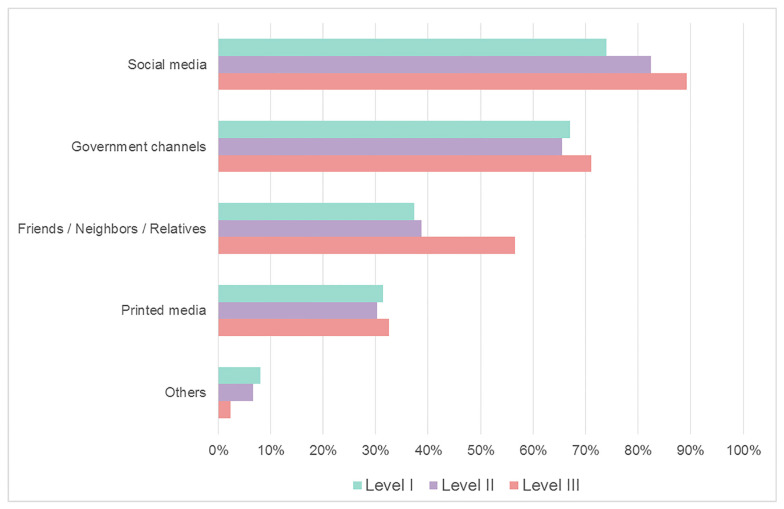
Information source of COVID-19 among three different lockdown levels. Level I, not restricted; level II, partially restricted; level III, restricted.

**Figure 3 ijerph-20-04944-f003:**
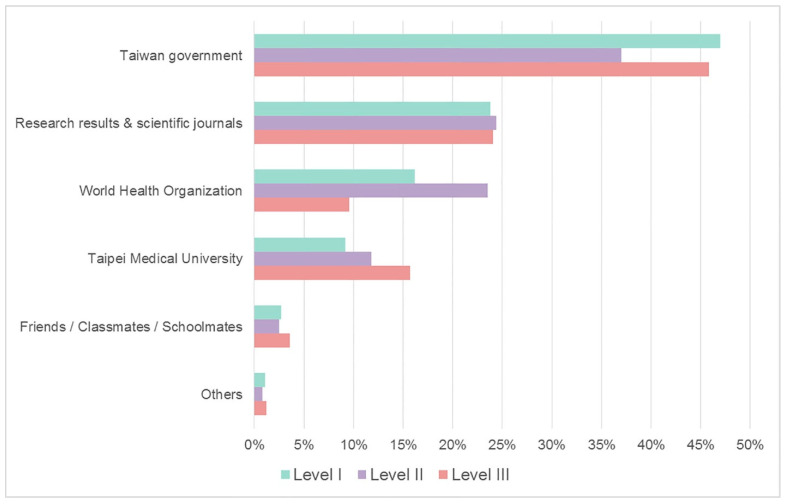
Reliable information of COVID-19 among three different lockdown levels. Level I, not restricted; level II, partially restricted; level III, restricted.

**Table 1 ijerph-20-04944-t001:** Lockdown policy of the Taiwan government for COVID-19.

Lockdown Level	I	II	III
Criteria	Imported cases resulting in isolated community transmission	Domestically transmitted cases from unknown sources	Three community clusters within a week or ten domestically transmitted cases from unknown sources in one day
Timing	2020.02.27~ 2021.05.10	2021.7.28~ 2022.02.28	2021.05.15~ 2021.07.27
Daily confirmed case number	<1	1–20	20–550
Daily death number	<1	1–10	10–40
Quarantine for suspected case	Mandatory	Mandatory	Mandatory
Border control	Highly restricted	Highly restricted	Highly restricted
Social distancing	Not mandatory	Semi-mandatory	Mandatory
Dining at restaurant	Not restricted	Partially restricted	Prohibited
Mass transportation	Normal frequency	Normal frequency	Reduced frequency
People-gathering leisure (e.g., exhibition, theater.)	Not restricted	Partially restricted	Prohibited
Religious activity	Not restricted	Partially restricted	Totally prohibited
Meeting	Not restricted	Partially restricted	Almost prohibited
Public wedding/funeral	Not restricted	Partially restricted	Prohibited
Wearing a mask	Mandatory only indoor	Mandatory in public areas	Mandatory all the time
Class in person	Not restricted	Partially restricted	Prohibited
Work	Not restricted	Work from home is encouraged	Work from home is highly recommended

Source of information: CECC.

**Table 2 ijerph-20-04944-t002:** Demographic information of participants.

Lockdown Level	I		II		III		*p*-Value *
	N = 185		N = 119		N = 83		
**Gender**							
Female	97	52.4%	68	57.1%	44	53.0%	0.481
Male	88	47.6%	50	42.0%	38	45.8%	
Others	0	0.0%	1	0.8%	1	1.2%	
**Age** (years old)							
<30 years old	72	38.9%	48	40.3%	33	39.8%	0.969
≥30 years old	113	61.1%	71	59.7%	50	60.2%	
**Home country**							
Indonesia	57	30.8%	34	28.6%	33	39.8%	0.776
Vietnam	55	29.7%	41	34.4%	29	34.9%	
Others	73	39.5%	44	37.0%	21	25.3%	
**Ethnicity**							
Asian	148	80.0%	102	85.7%	71	85.5%	0.844
Not Asian	37	20.0%	17	14.3%	12	14.5%	
**Religion**							
Islam	51	27.6%	23	19.3%	26	31.3%	0.377
Christianity	42	22.7%	24	20.2%	15	18.1%	
Buddhism	22	11.9%	21	17.6%	16	19.3%	
Others	70	37.8%	51	42.9%	26	31.3%	
**Marital status**							
Single	108	58.4%	75	63.0%	44	53.0%	0.455
Married	77	41.6%	44	37.0%	39	47.0%	
**Pursuing degree**							
Bachelor’s	0	0.0%	1	0.8%	1	1.2%	0.588
Master’s	59	31.9%	39	32.8%	25	30.1%	
Doctorate	126	68.1%	79	66.4%	57	68.7%	
**Living arrangement**							
Alone	56	30.3%	50	42.0%	24	28.9%	0.602
Not Alone	129	69.7%	69	58.0%	59	71.1%	
**Stay duration in Taiwan**							
<1 year	70	37.8%	47	39.5%	32	38.6%	0.959
≥1 year	115	62.2%	72	60.5%	51	61.4%	

* Chi-square test or Fisher’s exact test when appropriate.

**Table 3 ijerph-20-04944-t003:** Impacts of lockdown policy on the daily lives of international students.

Lockdown Level	I		II		III		*p*-Value *
	N = 185		N = 119		N = 83		
**Transportation, commute**							
Never/seldom	42	22.7%	20	16.8%	11	13.3%	
Sometimes	46	24.9%	29	24.4%	15	18.1%	
Always/often	97	52.4%	70	58.8%	57	68.7%	0.013
**Study**							
Never/seldom	51	27.6%	20	16.8%	7	8.4%	
Sometimes	60	32.4%	24	20.2%	16	19.3%	
Always/often	74	40.0%	75	63.0%	60	72.3%	<0.001
**Leisure**							
Never/seldom	39	21.1%	23	19.3%	11	13.3%	
Sometimes	68	36.8%	19	16.0%	14	16.9%	
Always/often	78	42.2%	77	64.7%	58	69.9%	<0.001
**Family life**							
Never/seldom	72	38.9%	37	31.1%	23	27.7%	
Sometimes	53	28.6%	23	19.3%	15	18.1%	
Always/often	60	32.4%	59	49.6%	45	54.2%	<0.001
**Diet behavior**							
Never/seldom	67	36.2%	26	21.8%	18	21.7%	
Sometimes	60	32.4%	39	32.8%	21	25.3%	
Always/often	58	31.4%	54	45.4%	44	53.0%	<0.001
**Income**							
Increased	4	2.2%	4	3.4%	2	2.4%	
No change	112	60.5%	68	57.1%	51	61.4%	
Decreased	69	37.3%	47	39.5%	30	36.1%	0.945

* Mantel–Haenszel linear-by-linear association.

## Data Availability

The datasets generated and/or analyzed during the current study are not publicly available due the regulation of the Institutional Review Board of Taipei Medical University but are available from the corresponding author on reasonable request.

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
