# Peer review of "Perceptions towards the COVID-19 Pandemic during Different Lockdown Levels among International Students in Taiwan"

_ijerph, 2023, doi:10.3390/ijerph20064944_

Round 1

Author Response

Point 1: Ambiguous statement:

Response 1: In this revised manuscript, we rephrased the sentence as your recommendation (page 1, lines 22~23).

Point 2: Comprehend the aim of the study with significance:

Response 2: We comprehended the aim of the study with significance as your recommendation (page 3, lines 71~75).

Point 3: Figure 1 -- improper graph style, need to be improved

Response 3: We improved the graph style of Figure 1 as your recommendation.

Point 4: Figure 2 -- Change the graph style.

Response 4: We improved the graph style of Figure 1 as your recommendation.

Reviewer 2 Report

The authors present a survey related to international student's perception of lockdown during the COVID-19 pandemic in 2021. It is interesting research with important descriptive results related to it. 

One point that is not present in the article is the dialogue between the context of Taiwan and other parts of the world during this period. What is the correlation between the levels of lockdown in Taiwan and the world situation of the pandemic in January to April 2021, May to July 2021, and August to December 2021? I suggest adding this information. 

Also, it is important to highlight that 2021 is the second year of lockdown in the world, it means people have already experienced this situation in 2020. Considering this, probably the authors would have another kind of results, if research was conducted in 2020. I think it would be important to highlight this information, because it can be related to the positive perceptions of the participants. 

In summary, I think a global context of the pandemic, and the contextualization in the second year of the lockdown experience, are important for a more general understanding and discussion of the research results. 

Congratulations for this article and the effort to bring discussion about difficult life in pandemic times.   

Author Response

Point 1: One point that is not present in the article is the dialogue between the context of Taiwan and other parts of the world during this period. What is the correlation between the levels of lockdown in Taiwan and the world situation of the pandemic in January to April 2021, May to July 2021, and August to December 2021? I suggest adding this information.

Response 1: We added the information as your suggestion (page 4, lines 134~160).

Point 2: Also, it is important to highlight that 2021 is the second year of lockdown in the world, it means people have already experienced this situation in 2020. Considering this, probably the authors would have another kind of results, if research was conducted in 2020. I think it would be important to highlight this information, because it can be related to the positive perceptions of the participants. 

Response 2: We highlighted the situation of our study period as your recommendation (page 9, lines 295~297).

Reviewer 3 Report

The present study assessed several aspects on perceptions towards the COVID-19 pandemic during different lockdown levels among international students in Taiwan. The study, in general, is well presented, guaranteeing the potential for replicability. However, there are major concerns and other minor concerns that I believe need to be addressed in order to improve the manuscript clarity and presentation.

Major concerns

First, since it was not detailed, I am concerned about the inclusion of questionnaires that were answered at the end of July (end of lockdown level III) and those that were answered in early August (beginning of lockdown level II). Was there enough time for students to notice significant impacts with so little time between data collection moments?

Second and finally, In the Figure 3, we can notice a lower percentage of response at level II for Taiwan government as we also see a higher percentage of response at the same level for World Health Organization. Is there an explanation for these observations? 

Minor concerns

Table 1 should have a note indicating the source of the information presented.

All abbreviations should be presented at the first time the expression which it represents appears in the text, for instance, “knowledge, attitudes, and practices” appears for the first time in Methods (page 5) and KAP abbreviation appears for the first time in Discussion (page 8). Check for al abbreviations.

In the item 3.1, I suggest the use of “participants” all times.

Also in the item 3.1, I suggest the deletion of several sentences that repeats what is already stated at Table 1 (see attachment).

As in Table 3, Table 2 should indicate with “*” and “Note” the test applied.

All figure’s captions should detail what do levels I, II and III mean, ensuring that the figures are independent from the text.

Every time a see the category Others, I kept myself wondering what its composition is.

I believe the study assesses information sources not information accesses.

You can find attached a commented file.

Author Response

Point 1: First, since it was not detailed, I am concerned about the inclusion of questionnaires that were answered at the end of July (end of lockdown level III) and those that were answered in early August (beginning of lockdown level II). Was there enough time for students to notice significant impacts with so little time between data collection moments?

Response 1: In this revised manuscript, we detailed the study period to show the time gap between lockdown level II and III (page 3, lines 81~83). We believe the time gap is sufficient.

Point 2: Second and finally, In the Figure 3, we can notice a lower percentage of response at level II for Taiwan government as we also see a higher percentage of response at the same level for World Health Organization. Is there an explanation for these observations? 

Response 2: We made a couple of explanations for these observations in this revised discussion section (page 9, lines 283~288).

Point 3: Table 1 should have a note indicating the source of the information presented.

Response 3: We provided a note indicating the source of the information in this revised Table 1 (page 2, line 50).

Point 4: All abbreviations should be presented at the first time the expression which it represents appears in the text, for instance, “knowledge, attitudes, and practices” appears for the first time in Methods (page 5) and KAP abbreviation appears for the first time in Discussion (page 8). Check for al abbreviations.

Response 4: Thank you for the correction. We checked for all abbreviations and revised them (page 3, line 96 & page 9, line 234).

Point 5: In the item 3.1, I suggest the use of “participants” all times.

Response 5: We replaced all samples with participants as your suggestion (page 4, lines 135, 136, 145, 146, and 155).

Point 6: Also in the item 3.1, I suggest the deletion of several sentences that repeats what is already stated at Table 1 (see attachment).

Response 6: We deleted some sentences as your suggestion (page 4, lines 136~160).

Point 7: As in Table 3, Table 2 should indicate with “*” and “Note” the test applied.

Response 7: We indicated “*” and “Note” as your recommendation (page 5, line 174).

Point 8: All figure’s captions should detail what do levels I, II and III mean, ensuring that the figures are independent from the text.

Response 8: We detailed what levels I, II, and III mean as your recommendation (page 6, lines 184&185; page 7 lines 210&211; page 8, lines 220&221).

Point 9: Every time a see the category Others, I kept myself wondering what its composition is.

Response 9: We provided the composition of “others” in this revised manuscript (page 7, lines 207~208; page 8, lines 217~218).

Point 10: I believe the study assesses information sources not information accesses.

Response 10: We replaced access with source in the revised text (page 7, lines 202 & 210).